# The Role of Nutritional Status, Gastrointestinal Peptides, and Endocannabinoids in the Prognosis and Treatment of Children with Cancer

**DOI:** 10.3390/ijms23095159

**Published:** 2022-05-05

**Authors:** Magdalena Schab, Szymon Skoczen

**Affiliations:** 1Department of Pediatric Oncology and Hematology, University Children’s Hospital of Krakow, 30-663 Krakow, Poland; magdalenaschab@interia.pl; 2Department of Pediatric Oncology and Hematology, Faculty of Medicine, Jagiellonian University Medical College, 30-663 Krakow, Poland

**Keywords:** cancer, children, nutritional status, endocannabinoid system, gastrointestinal peptides

## Abstract

Neoplastic diseases in children are the second most frequent cause of death among the young. It is estimated that 400,000 children worldwide will be diagnosed with cancer each year. The nutritional status at diagnosis is a prognostic indicator and influences the treatment tolerance. Both malnutrition and obesity increase the risk of mortality and complications during treatment. It is necessary to constantly search for new factors that impair the nutritional status. The endocannabinoid system (ECS) is a signaling system whose best-known function is regulating energy balance and food intake, but it also plays a role in pain control, embryogenesis, neurogenesis, learning, and the regulation of lipid and glucose metabolism. Its action is multidirectional, and its role is being discovered in an increasing number of diseases. In adults, cannabinoids have been shown to have anti-cancer properties against breast and pancreatic cancer, melanoma, lymphoma, and brain tumors. Data on the importance of both the endocannabinoid system and synthetic cannabinoids are lacking in children with cancer. This review highlights the role of nutritional status in the oncological treatment process, and describes the role of ECS and gastrointestinal peptides in regulating appetite. We also point to the need for research to evaluate the role of the endocannabinoid system in children with cancer, together with a prospective assessment of nutritional status during oncological treatment.

## 1. Introduction

Childhood cancer is a rarely occurring disease. However, it is the second most frequent cause of death among children. It is estimated that each year cancer will be diagnosed in 400,000 children worldwide [1]. The cure rate in developed countries reaches 80%, while in low-and middle-income countries it is only 30% [2]. The most frequently diagnosed childhood cancers include acute lymphoblastic leukemia, brain tumors, lymphomas, and solid tumors such as Wilms and neuroblastoma [1,2]. Chemotherapy and radiotherapy are necessary treatments but have many side effects. One of the elements evaluated at diagnosis, influenced by both the disease and the treatment, is nutritional status [3]. It should be assessed regularly during therapy using anthropometric measurements, biochemical tests, a detailed nutritional interview, and appropriate scales [4]. It is important to choose adequately sensitive measurement methods since the body mass index (BMI) does not distinguish between muscle and fat mass and may distort the results, especially in children with oedemas and solid tumors when the real body weight is masked, respectively, by water and the mass of tumor [5]. In oncology, the gold standard in assessing body composition is dual-energy x-ray absorptiometry (DXA). However, it has limited use in both low- and high-income countries [6]. A more affordable and cost-effective method of assessing nutritional status is the measurement of mid-upper arm circumference (MUAC), triceps skinfold thickness (TSFT), and arm muscle circumference (AMC), based on which it is possible to estimate the muscle and fat mass [4,6]. To assess body composition, the electrical bioimpedance analysis (BIA) can also be performed to evaluate the content of fat, lean tissue, and water [4]. Differences in sensitivity of detecting malnutrition with the use of individual measurement methods are presented by Lemos et al. [7], who assessed the nutritional status of 1154 children diagnosed with cancer based on BMI malnutrition was found in 10.8% at diagnosis, 27.3% based on TSFT, 24.5% based on MUAC, and 13.6% based on AMC.

Many factors contribute to the nutritional problems of children with cancer [8]. It is necessary to constantly search for new causes, one of which may be the altered functioning of the ECS. This system shows a multidirectional effect, and it has been best known for its regulation of energy balance. The role of the ECS is also described in an increasing number of diseases [9].

## 2. Nutritional Status in Children with Cancer

Nutritional status is the biochemical, structural, and functional state of the body resulting from the level of coverage of energy and nutrient requirements and the action of factors influencing absorption and metabolism [6]. In children diagnosed with cancer, the nutritional status is one of the prognostic values influencing treatment tolerance, quality of life, drug metabolism, and overall survival [4,10,11,12]. Malnutrition and overweight may occur at diagnosis or appear during and after oncological therapy but should not be treated as a normal condition at any stage of treatment. Despite the growing interest in this topic, malnutrition among cancer patients remains a serious problem, ranging between 40–90% in lower-middle-income countries and between 0–30% in high-income countries [3]. This value varies depending on the type and stage of cancer, phase of treatment, and assessment method [4]. In addition, patients with high-risk treatment protocols are more likely to be malnourished [13,14]. During oncological treatment, in some types of cancer, weight gain is also observed. According to Iniesta R. et al. [15], the prevalence of overnutrition ranged from 8% to 78%. It is believed that the first nutritional status assessment has to be performed at diagnosis and should be repeated regularly during treatment [4,13]. The most noticeable changes in the nutritional status occur in the first months after diagnosis [13,16]. In a prospective study, Paciarotti et al. [17] have shown that after the first three months of treatment, the content of adipose tissue in children with leukemia increased to 130% of the norm, while in children diagnosed with other cancer, it decreased from 78% at diagnosis to 70% of the norm after three months of treatment. Furthermore, weight loss of >5% in the first 3 months of treatment and >10% after 6 months were associated with poorer survival [12].

### 2.1. Nutritional Disorders in Children Diagnosed with Various Types of Cancer

#### 2.1.1. Leukemia

Leukemia is the most common childhood cancer [1]. Studies show that obesity and undernutrition are associated with worse survival in children with acute myeloid leukemia (AML) [18]. Moreover, a higher BMI at diagnosis was associated with worse event-free survival (EFS), poorer overall survival (OS), and higher mortality in children with AML and those with acute lymphoblastic leukemia (ALL) [19,20,21,22]. On the other hand, Orgel E. et al. [23] showed that BMI at diagnosis wasn’t as important as underweight and obesity occurring more than half of the time between induction and maintenance. Studies also indicate that obesity during induction increases the risk of persistent minimal residual disease (MRD) [24]. However, some studies do not confirm the relationship between BMI and EFS [25], MRD [26], and increased risk of recurrence [27,28]. Detailed studies are listed in Table 1.

It is also worth noting that children with leukemia are at risk of low muscle mass. Suzuki D et al. [29] have assessed the content of skeletal muscle in patients with ALL using CT imaging at the L3 level. Skeletal muscle loss was demonstrated in all patients after the induction, while sarcopenia developed in almost 30% of the study group [29].

#### 2.1.2. Solid Tumors

It is difficult to unequivocally define the prevalence of nutritional disorders in children with solid tumors due to the varied results of individual studies resulting from different methods and times of nutritional assessment, as well as a different selection of the study group [3]. It is believed that the risk of malnutrition in children with solid tumors is higher compared to other types of childhood cancers [13,30,31,32,33], although not all studies confirm this [34,35]. Studies indicate that both too high and low BMI is associated with worse OS [36,37], worse response to treatment [38], and a higher risk of toxicity [39] and complications [40]. On the other hand, Sharib JM. et al. [38] do not point to malnutrition as a factor associated with increased treatment toxicity. Tenardi R et al. [41] carried out a retrospective assessment of the nutritional status of children with Ewing sarcoma and osteosarcoma, where a high risk of experiencing extreme body changes was observed [41]. Burke [42] observed that the loss of >10% of body weight was associated with an increased number of days of hospitalization [42]. Lifson L.F. et al. [43] have shown that malnutrition in children with Wilms’ tumor reaches 66%, but no statistically significant relationship was found between nutritional status and survival [44,45]. In children with neuroblastoma, malnutrition is noticeable at diagnosis, and BMI decreases after 6 months of treatment, but no relationship has been found between BMI and survival [45]. In a prospective study by Avarnival et al. [46], it has been shown that the BMI of children with solid tumors decreased during the first 6 months of treatment and then gradually increased. It should be emphasized that the risk of malnutrition in children with solid tumors differs depending on the type of nutritional assessment method used. Children with solid tumors have a higher risk of malnutrition, regarding the MUAC, TSFT, and AMC indicators, compared to other methods [47]. This is caused by the tumor mass masking the real body weight, which impairs seemingly correct BMI measurements [47]. That is why it is so important to use measurements employing arm anthropometry, bioimpedance, or, if possible, DXA for these children. Furthermore, Joffe L. et al. [48] demonstrated by using single-slice T12-L1 images from routinely obtained chest CT scans that children with solid tumors lose skeletal muscle and fat in the early stages of therapy. Detailed studies are listed in Table 2.

#### 2.1.3. Central Nervous System (CNS) Tumors 

Brain tumors are the second most common cancer in children after leukemia [1]. Tsutsumi et al. [49] have estimated that at diagnosis, 6.7% of children with CNS tumors were malnourished, while 23.3% were overweight. Iniesta R. et al. [13] confirm that patients with brain tumors had the highest risk of being overweight and obese compared to other types of cancer. In a prospective study, Brinksma A. et al. [16] have shown that in the first three months of treatment, most children with brain tumors increased the rate of WFA, BMI, and had a higher content of adipose tissue and lower lean body mass compared to children with solid tumors and hematological neoplasms. Musiol K. et al. [50] observed that in children with brain tumors, BMI was the lowest during the maintenance and was significantly different compared to the control group. After the end of treatment, BMI increased significantly [50].

### 2.2. Bone Health in Children with Cancer

The effect of treatment on the bone mineral content is also a very important issue because 40% of bone mass is formed in childhood [51]. Children treated for cancer show worse bone formation and higher bone resorption [52]. Added to the causes of bone structure disorders should be anti-cancer drugs like methotrexate, ifosfamide, cyclosporine, doxorubicin, cisplatin, and glucocorticosteroids, as well as radiotherapy, bone marrow transplantation, and decreased physical activity [52]. Studies show that children during treatment and cancer survivors have a significantly poorer bone mineral density (BMD) and a higher risk of osteoporosis, osteopenia, and fractures (Table 3).

Children with cancer have a higher risk of developing nutritional disorders than healthy children. Children with solid tumors are believed to be at greater risk of malnutrition than children with leukemia, who are more likely to be overweight and obese. Both malnutrition and obesity have a negative impact on survival, the occurrence of treatment toxicity, and EFS. The nutritional status assessment should be carried out regularly during therapy, and the assessment methods should be adapted to the type of tumor and the child’s age. During treatment, it is also necessary to remember and counteract the long-term effects of anti-cancer therapy, e.g., the occurrence of osteopenia and osteoporosis in cancer survivors, because this group has a significantly increased risk of their occurrence.

## 3. Regulation of Appetite in Children with Cancer

### 3.1. Appetite Regulation

The regulation of appetite in humans is a complex mechanism influenced by many factors [69]. In the CNS, the hypothalamus is a key area influencing the regulation of appetite [69]. The starvation center is in the lateral hypothalamic area (LHA), and the satiety center is in the ventromedial hypothalamus (VMH) [69,70]. They are influenced by neuropeptides and hormonal signals from tissues and organs [71]. The integration of circulating signals of hunger and satiety takes place in the arcuate nucleus (ARC), within which there are two opposing neuronal systems [71]. The first of them is the orexigenic system, which stimulates appetite through neuropeptide Y (NPY) and agouti-related peptide (AgRP) [72]. The second one is the anorexigenic system suppressing appetite through a proopiomelanocortin (POMC) and the amphetamine-regulated transcript (CART) [72]. Then, signals are transmitted to the paraventricular hypothalamic nucleus (PVN), where they are integrated and modified [73]. The neurons of PVN send axons that secrete corticoliberin (CRH), thyroliberin (TRH), and oxytocin (OXT) [73]. The arcuate nucleus also communicates with VMH, which secretes mainly anorexigenic brain-derived neurotrophic factor (BDNF) [73], and LHA, which secretes the appetite-stimulating peptides orexin-A (OxA), orexin-B (OxB), and melanin-concentrating hormone (MCH) [71].

Peripheral appetite regulators include gastrointestinal and adipose tissue hormones reaching the CNS via the bloodstream [70,74] but not all of them can cross the blood-brain barrier [75]. Gut peptides also affect the brain via vagal afferent fibers [71]. Leptin is a peptide produced mainly by white adipose tissue, and its concentration in the body positively correlates with the BMI and the amount of adipose tissue [76]. The action of leptin in the human body is multidirectional, but best known is its participation in the regulation of hunger and satiety, where it stimulates the POMC/CART system and inhibits the secretion of NPY [69,71]. Ghrelin, a gastrointestinal hormone produced in humans mainly in the stomach by type A enteroendocrine cells, has the opposite effect as leptin [77]. Stimulation of the starvation center is the main function of ghrelin, which negatively correlates with the BMI and concentration of leptin and insulin [70,74,78]. The lesser known functions of ghrelin are modulation of taste sensation, glucose metabolism, and gut motility [70]. Ghrelin level positively correlates with the severity of anorexia and cancer cachexia in adult patients [79,80,81]. Another important regulator is insulin, a long-term signal of satiety that can cross the blood-brain barrier [71]. Insulin stimulates leptin synthesis and inhibits NPY/AgRP neurons [69,71]. Glucagon-like peptide 1 (GLP-1) inhibits gastric emptying and reduces appetite, and stimulates the pancreas to secrete insulin [82]. This hormone has an important role in glucose metabolism [82]. Another peptide inhibiting food intake is peptide tyrosine-tyrosine (PYY), secreted in the distal intestine after a meal, especially after a protein-rich one [83]. Cholecystokinin (CCK) is synthesized mainly in the duodenum and jejunum, but also in the CNS [84]. It suppresses appetite and stimulates intestinal motility and secretion of insulin, glucagon, and pancreatic enzymes [85].

Moreover, genetic factors also play a role in regulating the level of intestinal hormones and thus the appetite. This was confirmed by Czogała et al. [86], who assessed the importance of FTO and PLAG1 gene expression in the context of gastrointestinal and adipose tissue hormone levels. The results indicate that the level of FTO and PLAG1 expression positively correlated with the concentration of leptin in the blood serum and negatively with CCK and GLP-1, while the expression and methylation of FTO negatively correlated with the levels of resistin and visfatin [86].

### 3.2. The Causes of Appetite and Nutritional Status Disorders in Children with Cancer

There are many factors contributing to cancer malnutrition (Table 4 ), which in contrast to starvation-related malnutrition is not caused only by insufficient food intake [87,88].

During cancer treatment, the action of neurohormones, gastrointestinal, and adipose tissue hormones also may change [88]. Only a few studies have shown gastrointestinal peptide dysfunction in children with cancer, and most have been conducted in children with leukemia. Fayh et al. [90] carried out a systematic review, which showed a wide variation in results. Most of the included studies [90] looked at the concentration of leptin, but in only one research was a higher level of leptin observed in children with cancer [91], while the remaining studies found lower concentration or no difference compared with the control group [90]. Only two studies looked at ghrelin concentration, and one of them indicated lower ghrelin levels in children with cancer, which increased in later stages of treatment [91]. As the causes of the varied results, the authors indicate different types of cancers, treatments, and ages of children [90]. Agyrou et al. [92] (2019) presented an overview of research on ghrelin, leptin, and adiponectin levels in children with ALL. They noticed that in most studies, the leptin level was higher and the adiponectin level was lower at diagnosis [92]. Furthermore, Carvalho Gomes CC et al. [93] assessed the levels of appetite-regulating hormones in children with ALL during the induction at three-time points. Statistically significant changes have been observed in the level of ghrelin, which positively correlated with food consumption [93]. The concentration of leptin, insulin, and cortisol did not change significantly during the 28 days of the study [93]. Barbosa-Corte et al. [94] have shown that malnourished children with solid tumors and lymphomas have a lower leptin concentration than well-nourished children. Musial et al. [50] observed no statistically significant differences between leptin levels in children with CNS tumors and the control group and no correlation between leptin concentration and BMI. Statistically, an insignificant lower leptin level at diagnosis was observed in patients with brain tumors compared to the control group (7.04 vs. 16.38 ng/mL) and malnourished children [50]. Changes in gastrointestinal peptides have also been indicated by Skoczeń et al. [95], who observed that the concentration of CCK, ghrelin, and GLP-1 and the expression of their genes were significantly lower before bone transplant compared to 6 months after transplantation. Moreover, the concentrations of peptides in the test group were significantly lower than in the control group of healthy children [95]. The authors indicate that it may be caused by damage to the gastrointestinal mucosa, and the measurement of the concentrations of selected peptides may be a marker of gastrointestinal regeneration [95].

Appetite is also altered by disturbed gastrointestinal tract motility, taste, and smell disturbances, occurring at 45–84% for taste and 5–60% regarding the smell of adult cancer patients [8]. Cancer patients also exhibit increased protein catabolism and lipolysis, as well as inhibition of lipoprotein lipase [8,87]. The deterioration of the body condition leads to the development of cachexia characterized by unintentional weight loss, including muscle and fat tissue, marked weakness, dysfunctional immunity, decreased intestinal peristalsis, and abnormal functioning of the heart and other key organs and systems of the body [89].

Appetite regulation in humans is very complex and involves both CNS centers and peripheral factors. During oncological treatment, the action of hunger and satiety centers is disturbed by the action of proinflammatory cytokines, substances secreted by cancer, and metabolic disorders. In children with cancer, changes in the level of gastrointestinal peptides such as CCK, ghrelin, leptin, and GLP-1 are also observed. However, the results of individual studies are contradictory, and this issue requires further research.

## 4. The Role of Endocannabinoids System in Childhood Cancer

### 4.1. Physiology of ECS

ECS is a system of endogenous cannabinoids, receptors, enzymes, and transport proteins, discovered in the 90s [9,96]. It is found in mammals, other vertebrates, and some invertebrates [9,97]. In humans, it is already present in the embryo, while cannabinoid receptors in the brain are detected in the 14th week of fetal life [98]. Moreover, blocking CB1 receptors in mice in the first 24 h of life inhibited the suckling of milk [99]. Added to the best known and first discovered ECS ligands should be included anandamide (AEA) and 2-arachidoyl glycerol (2-AG) [100,101]. Both endocannabinoids are formed “on-demand” [102]. They are lipid derivatives of arachidonic acid (AA) belonging to omega-6 polyunsaturated fatty acids [9]. The precursor of AEA is N-acylphosphatidylethanolamine (NAPE), from which in the brain, kidneys, liver, lungs, spleen, and heart, anandamide and phosphatidic acid are formed using N-acyl phosphatidylethanolamine phospholipase D (NAPE-PLD) [9,97,103]. The 2-arachidonoyl glycerol is formed from diacylglycerol (DAG) using diacylglycerol lipase (DAGL) and phospholipase C [104,105,106,107]. Endocannabinoids have to leave the cell to fulfill their function, and due to their polar nature, the eCB membrane transporter must be involved. The termination of endocannabinoid signaling is intracellular [108]. AEA is mainly degraded by fatty acid amide hydrolase (FAAH) to ethanolamine and arachidonic acid, while 2-AG is hydrolyzed by monoacylglycerol lipase (MAGL) to arachidonic acid and glycerol [109]. It is worth noting that arachidonic acid formed due to hydrolysis is a substrate for the production of prostaglandins [9].

In the ECS, there are two main types of receptors—CB1 and CB2, which are G protein-coupled receptors (GPCRs) [96]. CB1 receptors are located primarily in the brain—most of them in the basal ganglia, cerebellum, hippocampus, and cortex [108]. They are also located in endocrine glands, thyroid and adrenal cells, ovaries, testes, uterus, and placenta, in the gastrointestinal tract, and in adipose tissue, where endocannabinoids activate lipoprotein lipase and fat deposition [110]. In addition, CB1 receptors are also found in the vagus nerve endings [110]. CB2 receptors are located mainly in cells of the immune system—on the surface of B lymphocytes, macrophages, monocytes, and NK cells. Moreover, they are also in the spleen, tonsils, and hematopoietic cells [111], and in the CNS they are located mainly in microglia [92]. Anandamide has a high affinity for the CB1 receptor and low affinity for CB2, while 2-AG can bind to both receptors [96].

### 4.2. The Role of the ECS in the Regulation of Appetite

The ECS in the human body works in a multidirectional way, and its role is still being investigated [9]. The best-known function of ECS is the regulation of energy balance and food consumption [9]. Other known functions of the ECS include pain control, thermogenesis, sleep cycle regulation, embryogenesis, neurogenesis, learning, and memory, as well as regulation of lipid and glucose metabolism [9,109,111].

In the appetite control process, CB1 receptors in appetite-regulating regions of the hypothalamus are involved, as well as CB1 receptors located in the limbic system, digestive tract, and adipose tissue [109]. It is worth noting that by the presence of these receptors in the limbic system, ECS takes part in the hedonic evaluation of food [70]. The ECS also decreases gastric acid secretion and gastrointestinal motility [112]. A study on an animal model confirms the role of endocannabinoids in regulating the energy balance. Mice with knockout CB1 receptors had lower body weight, were resistant to hyperphagia [113], and were insensitive to the action of leptin [114]. Activation of the CB1 receptors in OUN leads to increasing motivation for palatable foods and increasing odor sensitivity, which leads to a reduction of satiety feeling and increased food intake (Figure 1) [115,116]. ECS also interacts with gastrointestinal peptides. Activation of the ECS leads to ghrelin secretion, which increases appetite (Figure 1) [102]. This is a two-way action because the ECS system stimulates the secretion of ghrelin in the digestive system, while ghrelin stimulates the synthesis of 2-AG [116]. Furthermore, in CB1 knockout mice, ghrelin didn’t show an anorexigenic effect [117]. It has also been known that leptin levels negatively correlate with endocannabinoid concentration [110,118] (Figure 1). Activation of CB1 receptors also leads to an increase in insulin secretion, somatostatin, glucagon, and visfatin [110]. Furthermore, cholecystokinin reduces the expression of CB1 receptors [119].

A high-fat and high-calorie diet can also modulate the ECS. Tagliamonte et al. [120] have shown that overweight and obese people have lower plasma AEA levels after switching from the Western to the Mediterranean diet, possibly due to increased intake of polyunsaturated fatty acids and decreased consumption of saturated fatty acids. In another study, Tagliamonte et al. [121] have shown that fat and energy intake can influence the concentration of endocannabinoids, NAE, and NAPE.

### 4.3. The Role of ECS in Childhood Cancer

The potential role of ECS in the development and course of diseases is still being investigated. It is currently known that in children it plays a role in the pathogenesis of such diseases as immune thrombocytopenia, juvenile idiopathic arthritis, type 2 diabetes, inflammatory bowel disease, celiac disease, obesity, and inflammation of the nervous system [122]. There is no data in the literature on the role of ECS in oncological diseases in children. Most of the research is carried out among adults, where the anti-cancer properties of cannabinoids have been demonstrated in breast and pancreatic cancer, melanoma, lymphoma, and brain tumors [111]. Among children, there are single studies assessing the importance of the cannabinoid system and synthetic cannabinoids. Andradas [111] points out that most of the conducted studies concern acute lymphoblastic leukemia, which indicates that cannabinoids destroy cancer cells both in vivo and in vitro and that cannabinoids THC and CBD interact with vincristine, cytarabine, and doxorubicin in vitro [123,124]. It has also been shown that synthetic cannabinoids inhibit rhabdomyosarcoma growth [125] and reduce the viability and invasiveness of neuroblastoma cells [126]. Furthermore, synthetic cannabinoids induced cell cycle arrest of osteosarcoma cells [127]. In the case of brain tumors, it has been shown that in the group of children with low-grade gliomas, the level of CB1 expression was a predictor of spontaneous involution [128]. Study details are listed in Table 5.

The role of ECS in children’s oncology is still little known. In vitro studies indicate the anti-cancer effects of cannabinoids on leukemia, neuroblastoma, rhabdomyosarcoma, and osteosarcoma cells. Furthermore, cannabinoids can enhance the toxic effects of drugs. Another interesting issue is the interaction of endocannabinoids with gastrointestinal peptides. Endocannabinoids correlate positively with ghrelin secretion and negatively with leptin secretion. This topic also requires future research, especially in children with cancer.

## 5. Conclusions


both underweight and obesity in children with cancer are associated with adverse outcomes, poorer EFS and OS;children with cancer and cancer survivors are at risk of developing osteoporosis and osteopenia;the literature lacks studies prospectively assessing the nutritional status over the entire period of the oncological treatment as well as studies on the importance of gastrointestinal hormones and the endocannabinoid system in this group of patients;the current state of knowledge allows only to suspect the existence of a relationship between nutritional status, gastrointestinal peptides and endocannabinoids;this issue requires further research, and it is important not only due to the possible impact on the nutritional status but also due to the multidirectional action of the ECS, which may be important in future oncological therapies;


## Figures and Tables

**Figure 1 ijms-23-05159-f001:**
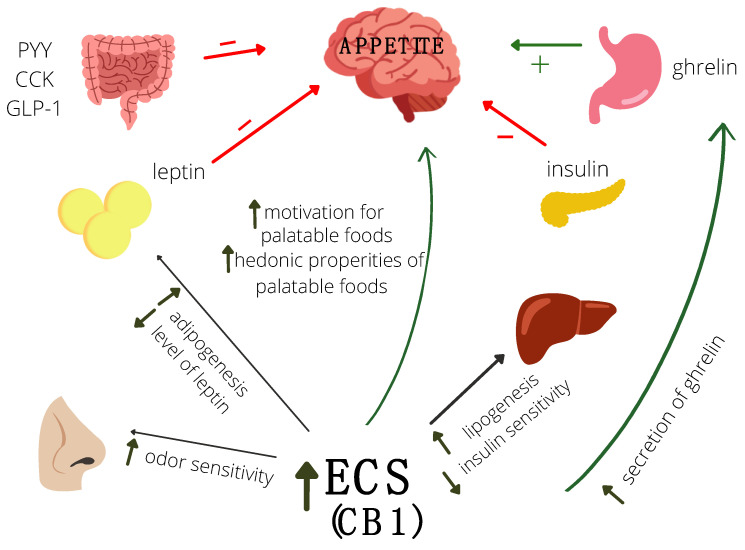
The role of endocannabinoid system (ECS) and gastrointestinal peptides in the regulation of appetite.

**Table 1 ijms-23-05159-t001:** Characteristic of studies that assess nutritional status of children with leukemia.

Type of Cancer	Patients (*n*)	Assesment Method	Nutritional Status at Diagnose	Outcome, Effect on Treatment	Age	Study Design	Year, Reference
**AML**	768	BMI ^a^	10.9% underweight 14.8% patients overweight	underweight patients had poorer survival (HR: 1.85; 95% confidence interval CI: 1.19–2.87; *p* = 0.006) and higher risk of treatment-related mortality (HR, 2.66; 95% CI: 1.38–5.11; *p* = 0.003) compared with middleweight patients; overweight patients had poorer survival (HR, 1.88; 95% CI: 1.25–2.83; *p* = 0.002), and higher risk of treatment-related mortality (HR: 3.49; 95% CI: 1.99–6.10; *p* < 0.001) and had higher leukocyte level (*p* = 0.001) compared with middleweight patients;	1–20	retrospective study	2005 [18]
**AML, ALL**	11,602	BMI ^b^	-	ALL—patients with BMI ≥ 85th percentile had poorer EFS (RR: 1.35; 95% CI: 1.20, 1.51) and increased mortality (RR: 1.31; 95% CI: 1.09, 1.58) compared with patients with BMI < 85th; AML—patients with BMI ≥ 85th percentile had poorer EFS (RR: 1.36; 95% CI: 1.16, 1.60) and OS (and RR: 1.56; 95% CI: 1.32, 1.86) than patients with BMI < 85th;	0–21	meta-analysis	2016 [19]
**AML, ALL**	181	BMI ^c^	28.8% overweight/obese 71.2% non-overweight,	statistically significant association between mortality and obesity in unadjusted models (imputed: HR = 2.54, 95% CI = 1.15–5.60, *p* = 0.021; complete set: HR = 2.72, 95% CI = 1.26–5.91, *p* = 0.011) in overweight/obese patients ≥ 10 years was observed trend towards increased risk of relapse (HR = 2.89, 95% CI = 0.89–9.36, *p* = 0.08) (age- and sex-adjusted analysis)	2–20	retrospective analysis	2018 [20]
**AML, ALL**	13,921	BMI	-	obesity at diagnosis was associated with increased risk of mortality (overall survival: HR =1.30, 95% CI = 1.16–1.46, *p* < 0.001, and event-free survival: HR = 1.46, 95% CI = 1.29–1.64, *p* < 0.001)		systematic review	2016 [21]
**ALL**	4260	BMI ^d^	8% obese 92% non obese	5-year event-free survival rate was higher in nonobese patients compared with obese 77% ± 0.6% vs. 72% ± 2.4% (*p* = 0.02); obesity patients had higher risk of events HR: 1.36 (95% CI, 1.04 to 1.77; *p* = 0.021) and relapses HR: 1.29 (95% CI, 1.02 to 1.56; *p* = 0.04); study cohort: obese patients ≥ 10 years had higher HR of events 1.5 (95% CI, 1.1 to 2.1; *p* = 0.009) and relapses 1.5 (95% CI, 1.2 to 2.1; *p* =0.013) compared to non-obese patients; verification cohort: obese patients ≥ 10 years, had higher HR of events—1.42 (95% CI, 1.03 to 1.96; *p* = 0.032) and relapses 1.65 (95% CI, 1.13 to 2.41; *p* = 0.009);	3–20	retrospective cohort study	2007 [22]
**ALL**	2008	BMI ^e^	5.8% underweight, 13.9% obese	obesity and undernutrition at diagnosis were associated with poorer EFS (HR = 1.40; 95% CI, 1.13 to 1.73 and HR = 1.33; 95% CI, 0.97 to 1.83, respectively; global *p* = 0.005); obesity and undernutrition at diagnosis and for ≥50% of the time between end of induction and start of maintenance were associated with poorer EFS (HR = 1.43; 95% CI, 1.04 to 1.96 and HR = 2.3; 95% CI, 1.46 to 3.63, respectively *p* < 0.001); obese patients were more likely to had hepatic and pancreatic toxicities (OR = 1.32; 95% CI, 1.15 to 1.51 and OR, 1.53; 95% CI, 1.22 to 1.92, respectively); underweight patients were more likely to had pulmonary toxicity and fungal infections (OR = 2.07; 95% CI, 1.31 to 3.29; *p* = 0.003 and OR = 2.24 95% CI, 1.51 to 3.32; *p* = 0.001, respectively);	1–20	retrospective cohort study	2014 [23]
**ALL**	198	BMI ^f^	20.7% obese, 15.2% overweight, 64% patients were “lean”	obesity at diagnosis was associated with higher risk of MRD positive at the end of induction (OR = 2.57; 95% CI = 1.19 to 5.54; *p* = 0.016) compared with non-obese patients; obesity and overweight were associated with poorer EFS irrespective of end-induction MRD (*p* = 0.012);	1–21	retrospective cohort study	2017 [24]
**ALL**	621	BMI ^g^	16.4% underweight 10.3% at risk of overweight 8.9% overweight	there were no statistical differences between BMI groups in overall survival (*p* = 0.533), event-free survival (*p* = 0.722), and cumulative incidence of relapse (*p* = 0.862);	>1 year	retrospective study	2008 [25]
**ALL**	373	BMI ^h^	7% underweight 12.1% overweight, 15.5% obese	no association between BMI and OR, EFS, cumulative incidence of relapse/ refractory disease (CIR) and MRD (*p* > 0.05);	>2	retrospective study	2017 [26]
**ALL**	172	BMI ^h,i^	CDC: 14.9% underweight, 14.9% overweight, 11.8% obese WHO: 3.5% patients were assigned to the severely wasted or wasted group, 22.1% at risk of overweight, 7.0% overweight, 2.3% obese.	no association between BMI determined by CDC or WHO criteria at diagnosis and DFS and OS;	0.5–15.5 (5)	observational retrospective study	2021 [28]

**DFS**—disease-free survival, **OS**—overall survival, **ALL**—acute lymphoblastic leukemia, **AML**—acute myeloid leukemia, **EFS**—event free survival, **BMI**—body mass index. **a**—BMI defined as: underweight BMI ≤ 10th percentile, overweight BMI ≥ 95th percentile, middleweight BMI 11th–94th percentiles, **b**—BMI defined as higher and lower: higher BMI defined as BMI ≥ 85% or lower defined BMI < 85%, **c**—BMI defined as overweight/obese ≥ 85th percentile or non-overweight < 85th percentile, **d**—obesity defined as BMI ≥ 95th percentile, **e**—underweight defined as BMI < 5th percentiles, obese BMI ≥ 95th percentile, normal weight or overweight BMI 5–95th percentiles, **f**—obese defined as BMI ≥ 95th percentile, overweight BMI 85–94th percentiles, “lean” BMI < 85th percentiles, **g**—underweight defined as BMI ≤ 10th percentile, normal weight—BMI 10–85th percentiles, at risk of overweight BMI ≥ 85th and <95th percentile, overweight BMI ≥ 95th percentile, **h**—CDC criteria: underweight defined as BMI < 5th percentiles, normal weight BMI 5–84.9th percentiles, overweight BMI ≥ 85th and <95th percentile, obese BMI ≥ 95th percentile, **i**—BMI defined as WHO criteria Z-score, normal weight −1.9−0.9, wasted-severely wasted < −2, risk of overweight 1–1.9, overweight-obesity ≥ 2.

**Table 2 ijms-23-05159-t002:** Characteristic of studies that assess the nutritional status of children with solid tumors.

Type of Cancer	Patients	Assessment Method	Nutritional Status at Diagnose	Outcome, Main Findings	Age (Years)	Study Design	Year, References
**mixed**	82	BMI ^a^, MUAC, TSF, BIA	all patients: 13% undernutrition, 7% overweight, 15% obese, (BMI) solid tumors: 17% undernutrition, 8.5% obesity, (BMI)	undernutrition at diagnosis was associated with risk of event defined as relapse, death or becoming palliative (19.901; *p* < 0.001); overnutrition at diagnosis was not associated with risk o event (*p* = 0.03) patients with solid tumors had the highest prevalence of undernutrition at diagnosis compared to haematological malignances and brain tumors (*p* < 0.05) (17% by BMI and 18% by TSF) after 3 first months of treatment BMI (*p* < 0.001) and FM (BIA) (*p* < 0.05) increased, whilst FFM (BIA) (*p* < 0.05) significantly decreased during this time high-treatment risk was associated with undernutrition during the first three months of treatment [*p* = 0.04; 95% CI (−16.8 to (−0.4)]	<18	prospective cohort study	2019 [13]
**solid tumors, hematological malignances**	74	weight, height, BMI, MUAC, TSF, SSF, dietary intake	patients with solid tumors: 29.7% severely underweight, 10.8% stunted, 8.1% lean, and 45.9% wasted	patients with solid tumors had a significantly lower mean BMI (*p* < 0.05), TSF (*p* < 0.01), SSF (*p* < 0.01), and sums of TSF and SSF (*p* < 0.01) compared with patients with hematological malignancies hematological malignances patients had higher intake of energy, protein, carbohydrate, vitamin A, and niacin than children with solid tumors (*p* < 0.05) children with solid tumors had more eating problems (loss of appetite, nausea, and vomiting) than children with hematological malignances (*p* < 0.05)	3–15	cross-sectional study	2012 [30]
**solid tumors, hematological malignances**	74	BMI ^b^, MUAC, TSFT, STRONGkid, PYMS	all patients: 12.3% undernutrition, 6.8% overnutrition solid tumors: 16.7% undernutrition, 5.6% overnutrition	no statistical differences between prevalence of undernutrition in solid tumor patients at baseline (16.7%) and hematologic malignances (10.9%) (*p* = 0.869) after 6 months of treatment, the prevalence of undernutrition decreased to 6.7% in the overall study population and 9.1% in patients with solid tumors; STRONGkids and PYMS revealed a high risk for malnutrition at diagnosis in 30.4% and 39.4% of patients with hematologic malignancies, and in 22.2% and 27.8% of patients with solid tumors, respectively	1–18	prospective observational cohort study	2019 [31]
**mixed**	366	BMI ^c^, MUAC	BMI at diagnosis: 15% undernutrition, 18% overweightMUAC at diagnosis: 23% undernutrition, 6% overweight	in children with solid tumors MUAC identified more undernourished patients (23%) compared with BMI (15%), while BMI identified more overweight children with solid tumors (18%) compared to MUAC (6%) (*p* = 0.001) no significant difference in the 10-year overall survival by the malnutrition measured by BMI (*p* = 0.1507) or MUAC (*p* = 0.8135) the highest prevalence of undernutrition measuring by MUAC was in the solid tumor group (23%) compared with hematological and CNS cancers (11,5%, 8,6%, respectively) the highest prevalence of undernutrition measuring by BMI was in the CNS tumor group (20.7%) compared with solid tumor and CNS cancers (15%, 8.2%, respectively)	3 months–18 years	retrospective cross-sectional study	2021 [32]
**solid tumors, hematological malignances**	127	BMI ^d,^ MUAC, TSFT, AMC	solid tumors group: undernourished was29.4% by BMI, 45.6% by TSFT, 44.1% by MUAC, 33.8% by AMC	Patients with solid tumors had higher prevalence of malnutrition compared with hematological patients group, measured by BMI-z-score (29.4% vs. 6.8%, *p* < 0.05), MUAC (44.1% vs. 25.4%, *p* < 0.05), AMC (33.8% vs. 10.2%, *p* < 0.05) Higher percentages of deficits were shown by TSFT and MUAC than by z-score/BMI	1.08–24.58	prospective study	2005 [33]
**solid tumors, hematological malignances**	1154	TSFT, MUAC, AMC, BMI ^e^, percentage weight loss	10.85% < adequate BMI, 20% > adequate BMI	no significant difference in the prevalence of malnutrition was observed between patients with solid tumors and hematological malignances in solid tumor group MUAC, TSFT identified more malnourished children compared with BMI and AMC (25.78%, 26.38% vs. 12.2%, 14.33%)	0–19	transversal observational study	2014 [34]
**Ewing sarcoma**	50	BMI ^f^	16% underweight, 20% obese	abnormal BMI (underweight and obese) associated with poorer histologic response to treatment compared with patients with normal BMI (OR = 4.64, 95% CI 1.12–19.14 *p* = 0.034) and worse OS (HR = 3.46, 95% CI 1.19–9.99 *p* = 0.022) abnormal BMI not statistically significant associated with EFS	9.7–20.1	retrospective study	2015 [37]
**Ewing sarcoma**	142	BMI	-	BMI not associated with TRT	<21	retrospective study	2012 [38]
**osteosarcoma**	498	BMI ^g^	14.7% low BMI, 8.6% high BMI	patients with high BMI had increased risk of arterial thrombosis (OR = 9.4, *p* = 0.03) patients with low BMI had increased risk of wound infection or slough (OR = 2.0, *p* = 0.07)	3.7–30	retrospective study	2011 [40]
**osteosarcoma**	710	BMI ^f^	10.4% low BMI, 26.6% high BMI	high BMI associated with renal toxicity in course 2 of therapy (OR = 2.7, 95% CI 1.2–6.4, *p* = 0.01), poorer OAS at 5 years compared to patients with normal BMI—69.7% vs. 80.5% (HR = 1.6, 95% CI 1.1–2.2, *p* = 0.005) and worse EFS at 3 years 66.2% vs. 75.5% (HR = 1.3 95% CI 0.9–1.8, *p* = 0.05)	2–20	retrospective study	2013 [39]
**osteosarcoma, Ewing sarcoma**	139	BMI ^h^	at diagnosis: Ewing sarcoma 12.9% underweight, 8.1% overweight, 3.2% obese osteosarcoma 7.8% underweight, 10.4% overweight, 11.7% obese	patients with Ewing sarcoma or osteosarcoma are at a high risk of malnutrition, including extreme changes in body weight during therapy nutritional status not associated with outcome	1–27	retrospective study	2012 [41]
**rhabdomyosarcoma**	468	BMI ^h^	9,83% underweight, 12.82% overweight, 11.54% obese	lost weight more than 10% from baseline associated with increased toxicities and increased number of days hospitalized when compared with patients who lost no more than 5% from baseline (OR = 1.24, 95% CI 1.00 – 1.54 *p* = 0.0463) BMI not associated with infection rate	2–20	retrospective study	2013 [42]
**Wilms’ tumor**	76	weight, height, MUAC, TSFT, BMI ^i^	BMI: 17.33% underweight mild, 6.67% underweight moderate, 5.33% underweight severe	malnutrition was not associated with poor outcome stage of disease was not significantly associated with nutritional status	0.9-12.4	prospective study	2016 [43]
**Wilms’ tumor**	1532	weight-for-age ^j^, BMI ^k,j^	<2 years old (*n* = 493) 15.8% low WFA, 15.2% high WFA ≥2 years old (*n* = 1039) 15% low BMI, 13% high BMI	no association between weight-for-age or BMI-for-age and EFS (*p* = 0.28)	<2 years, >2 years	retrospective study	2009 [44]
**Neuroblastoma**	154	BMI ^k^	24.0% underweight, 11.6% overweight	no statistically significant association between BMI and OS (*p* > 0.05) after 6 months of treatment, the BMI decreased in all children except patients with 4s disease (*p* < 0.01) and increase from baseline by 24 months (*p* = 0.007)	0–10.6	retrospective study	2015 [45]
**Mixed**	139	BMI ^l^	28% undernourished	patients with solid tumors, AML/CML, and CNS tumors were more likely to be malnourished compared to patients with ALL or lymphomas (RR 2.3; 95% CI, 1.3–3.9; *p* < 0.001)	2–16	retrospective study	2019 [46]
**Mixed**	100	BMI ^d^, MUAC, TSFT, AMC, albumin level	Malnourished was 37% of children by weight for age, 20% by height for age, 33% by BMI, 50% by TSFT, 39% by MUAC, 42% by AMC, 28% by albumin level	the overall prevalence of malnutrition was higher using arm anthropometry like MUAC and TSFT compared to measurement parameters like W/H z-scores or BMI	<18	prospective study	2008 [47]

**MUAC**—mid-upper arm circumference, **TSF**—triceps skinfold, **SSF**—subscapular skinfold, **TSFT**—triceps skin fold thickness, **AMC**—arm muscle circumference, **PYMS**—Paediatric Yorkhill Malnutrition Score, **a**—undernutrition defined as BMI < 2.3rd centile; −2 SD, overweight as BMI ≥ 85th < 95th centile; ≥+1.05 SD < 1.63 SD, obese as BMI ≥ 95th centile; ≥1.63 SD, healthy weight BMI > 2.3rd to <85th centile, **b**— BMI for age z-score: undernutrition z-score < −2 SD, normal nutritional status z-score ≥ −2, ≤2 SD), overnutrition z-score > 2 SD, **c**—BMI for age z-score: undernutrition z-score < −2, normal z-score ≥ −2 and ≤+2 for children 5 years old and younger; normal and risk of overweight z-score ≥ −2 and ≤1 for children over 5 years old, overweight z-score > +2 for children 5 years old and under, overweight, obesity and severe obesity z-score > +1 for children over 5 years old, **d**—BMI defined as WHO criteria z-score: normal weight ≥ −2 SD and <1 SD, risk of overweight ≥ 1 SD and <2 SD, overweight-obesity ≥ 2 SD, wasted-severely wasted < −2 SD, **e**—BMI defined as below adequate z-score < −2 SD, adequate z-score ≥ −2 SD and ≤+1 SD, above adequate z-score > +1 SD, **f**—BMI defined as low <5th percentile, normal 5–85th percentile, overweight 85–95th percentile, obese > 95th percentile, **g**—low BMI defined as ≤10th percentile, middle BMI 11–94th percentile, high BMI ≥ 95th percentile, **h**—adequately nourished defined as BMI ≥ 5th to <85th percentiles, underweight < 5th percentile, overweight > 85–95th percentile, obese > 95th percentile, **i**—underweight mild defined as BMI < −1 SD, underweight moderate BMI < −2 SD, underweight severe < - 3 SD, **j**—<2 years old: low Weight For Age (WFA) index < 10th percentile, high WFA > 90th percentile, ≥2 years old low BMI defined <10th percentile, high BMI > 90th percentile, **k**—underweight defined as BMI < 15th percentiles, normal weight 15–85th percentiles, overweight > 85th percentile, **l**—ISO-BMI, underweight defined as < 17 kg/m^2^, healthy weight 17–24.9 kg/m^2^, overweight 25–29.9 kg/m^2^, obese ≥ 30 kg/m^2^.

**Table 3 ijms-23-05159-t003:** Characteristic of studies that assess bone health of children with cancer and survivors.

Cancer Type	Patients (*n*)/Control Group (*n*)	Assessment Method	Main Findings	Year, References
**ALL**	28/28	lumbar and total areal BMD, %FM, DXA, physical activity (accelerometer and questionnaire)	lumbar BMDvol in ALL survivors was significantly lower than in controls (*p* < 0.01); weekly activity score (by questionnaire) was significantly lower in the ALL group than in the control group (*p* < 0.05); male gender, low activity levels, intravenous high dose of methotrexate were associated with low lumbar BMDvol;	2002 [53]
**ALL and solid tumors**	28 (10 with ALL, 18 with solid tumors)	lumbar spine and femoral neck BMD, DXA, biochemical tests	femoral BMD and apparent volumetric density were decreased 1 year after diagnosis (*p <* 0.01]; the markers of bone formation-PICP and OC were significantly decreased at diagnosis, and by the end of the study were normalized; marker of bone resorption (type I collagen carboxyl-terminal telopeptide) was significantly increased at the end of the study; levels of 25-hydroxyvitamin D, 1,25-dihydroxyvitamin D, IGF- binding protein-3 were significantly decreased during the study;	1999 [54]
**ALL**	103	DXA, BMD	33% of patients had low BMD, and 4.9% of patients had osteoporosis;	2017 [55]
**ALL**	122	incidence of fractures, ON, bone pain	the relative rate of fractures was 2.03 (95% confidence interval 1.15–3.57), with greatest rates in children < 5 years; the 5-year incidence of fractures, ON, and isolated bone pain was 13.5%, 12.1%, and 12.3%, respectively;	2007 [56]
**ALL**	155	BMD, lateral thoracolumbar spine radiographs, incident vertebral fractures	16% of children with ALL developed incident vertebral fractures 12 months after the initiation of therapy; no association between glucocorticoid or methotrexate dose and incident vertebral fracture;	2012 [57]
**ALL**	186	BMD, lateral thoracolumbar spine radiograph, bone age	children with grade 1 or higher vertebral compression had lower lumbar spine areal BMD Z-scores compared with children without (*p* < 0.001); lumbar spine BMD and back pain were associated with increased odds of fracture;	2009 [58]
**ALL**	124	DXA	at diagnosis: 30% had osteopenia, 11% had osteoporosis; during therapy-39.5% had osteopenia, 8% had osteoporosis; 18.5% patients developed fractures; predictors of fracture: dexamethasone therapy (*p* = 0.01), lower LS-BMD (*p* = 0.01);	2011 [59]
**ALL, lymphoma**	ALL–22 lymphoma–4	DXA	BMC and areal BMD were significantly lower than in healthy controls; no association between BMT and whole-body bone mass;	2000 [60]
**osteosarcoma, Ewing’s sarcoma**	Ewing’s sarcoma-18 osteosarcoma-25	DXA, BMD, fracture rate	58% had BMD reduction; 16% had fractures;	2012 [61]
**osteosarcoma**	40/55	DXA	47.5% had osteoporosis, 30.0% had osteopenia; risk factor of osteoporosis: young age at diagnosis, male sex, ow lean mass;	2013 [62]
**adult survivors of childhood brain tumors**	74	DXA, biochemical tests	BMD was decreased in all measurement sites; male sex was associated with low BMD (*p* < 0.05); FSH and LH were negatively associated with BMD in women (*p* < 0.05);	2018 [63]
**high-risk neuroblastoma**	21/20	BMD, DXA, spinal magnetic resonance imaging	86% survivors had at least one skeletal adverse event; 38% had a severe complication;	2017 [64]
**ALL**	39	DXA, BMD	23.1% had osteopenia and 7.7% had osteoporosis;	2019 [65]
**ALL**	122		18% of survivors displayed osteopathologies; 77% had impaired bone health (at least one pathological screening parameter); 15% had vitamin D deficiency;	2020 [66]
**osteosarcoma**	9/8	DXA, BMD of the lumbar spine and femur neck	44% had decreased lumbar spine BMD (*p* = 0.024); 78% had decreased femur neck BMD (*p* = 0.023);	2015 [67]
**osteosarcoma**	48 long term survivors > 10 years	BMD of the lumbar spine and proximal femur, DXA, biochemical tests	association between C-telopeptides with the BMD (*p* = 0.04)	2003 [68]

**BMD**—bone mineral density, **DXA**—dual-energy x-ray absorptiometry, **ALL**—acute lymphoblastic leukemia, **BMDvol**—volumetric bone mineral density, **ON**—osteonecrosis, **BMT**—bone marrow transplantation, **OC**—osteocalcin, **PICP**—type I collagen carboxylterminal propeptid.

**Table 4 ijms-23-05159-t004:** Causes of appetite loss in cancer patients [87,88,89].

Factors	Description
hormonal imbalance	gastrointestinal hormones adipose tissue hormones neurohormones
proinflammatory cytokines	IL-1 IL-6 TNF-α IFN-y
substances secreted by tumor	PIF PMF LMF
metabolism changes	proteolysis ↑ lipolysis ↑ glycolysis ↑
side effects of treatment	taste and smell dysfunction nausea, vomiting impaired intestinal motility oral mucositis

IL-1—interleukin 1, IL-6—interleukin 6, INF-y—interferon gamma, LMF—lipid mobilizing factor, PIF—protein inducing factor, TNF-α—tumor necrosis factor, PMF—protein mobilizing factor

**Table 5 ijms-23-05159-t005:** Characteristic of studies that assess cannabinoid anti-cancer potential in children.

Type of Cancer	Cannabinoid/ECS Element	Details	Main Findings	Year, References
**leukemia**	CBD, CBG, THC	human cancer cell lines CEM (acute lymphocytic leukemia) and HL60 (promyelocytic leukemia)	combination of endocannabinoids (especially with CBD) has a greater anti-cancer response compared with the use of cannabinoids separately; combination of endocannabinoids worked synergistically with vincristine and cytarabine; greater induction apoptosis was observed when cannabinoids were used after anti-cancer drugs;	2017 [123]
**leukemia**	THC	leukemic cell lines CEM (lymphoblastic), HL60 (promyelocytic), and MOLT4 (lymphoblastic)	THC significantly strengthened the action of cytarabine, doxorubicin, and vincristine in reducing cell number and viability; THC makes leukemic cells more sensitive to the cytotoxic effects of chemotherapy;	2008 [124]
**rhabdomyosarcoma**	HU210, Met-F-AEA AM251, THC	Rh4, Rh28 (translocation positive rhabdomyosarcoma cells) RMS13, RD, and MRC-5 (lung fibroblast cells)	HU210, THC, and Met-F-AEA have proapoptotic effects on tposRMS cells through the CB1 receptor; HU210, THC, and Met-F-AEA reduce viability through up-regulation of transcription factor p8;	2009 [125]
**neuroblastoma**	CBD, THC	SK-N-SH	CBD and THC have antitumourigenic activity in vitro and decreased growth of tumors in vivo; CBD was more active than THC; CBD induces apoptosis and increases caspase-3 levels in the SK-N-SH neuroblastoma cell, and reduced the viability and invasiveness of tumor cells in vitro;	2016 [126]
**neuroblastoma**	AM404 (ECS modulator)	SK-N-SH	AM404 inhibits NFAT and NF-κB transcriptional activity by CB1- and TRPV1-independent mechanism; AM404 inhibits MMP-1, -3, and -7 expression and cell migration;	2015 [129]
**osteosarcoma**	WIN, ANA, MethANA, 3-MA	MG63, Saos-2	WIN decreased cell number and morphological alterations, with no association with induction of cell death; WIN induced G2/M cell cycle arrest;	2014 [127]
**low-grade glioma**	CB1 receptor	33 sample LGG	in LGG pediatric tumors which remained stable or underwent spontaneous involution observed high CNR1 expression at diagnosis	2016 [128]
**leukemia**	CP55940	Jurkat clone E6-1 (T-ALL), PBL	CP55940 induced production of ROS and apoptosis in Jurkat cells, but not in PBL; Mechanism of cell death in Jurkat is CBR-independent;	2020 [130]

**CBD**—Cannabidiol, **CBG**—cannabigerol, **THC**—Δ9-tetra- hydrocannabinol.

## Data Availability

Not applicable.

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
