# Peer review of "The Role of Nutritional Status, Gastrointestinal Peptides, and Endocannabinoids in the Prognosis and Treatment of Children with Cancer"

_ijms, 2022, doi:10.3390/ijms23095159_

Round 1

Reviewer 1 Report

The article is dealing with highly important issues of roles of the nutritional status, gastrointestinal peptides, and endocannabinoids in children with cancer. The text is well written, concise, and precise allowing readers to easily follow the presented ideas and facts. To be even more clear, I suggest authors add tables in which previously published studies regarding these issues are systematically shown.

Author Response

Dear Reviewer,

Reviewer 2 Report

The authors of the manuscript addressed a very important topic of the nutritional status of children with cancer, its impact on the course of treatment and prognosis. The manuscript is written carefully, nevertheless, to improve the value and citation of the publication, I propose the following change

  1. Title:The role of nutritional status, gastrointestinal peptides, and endocannabinoids in children with cancer” – specify: role of nutritional … in children with cancer - in what? treatment? prognosis? etc.
  2. Chapter 2 is chaotic. Describing in one sentence what who has discovered is not enough. Try to organize the information according to the issues and consider presenting the test results in the table, including, for example, the type of cancer, nutritional status, effect on treatment, prognosis, references. Rewriting the values of statistics from research adds little. Try to present the results and necessarily add conclusions at the end of each chapter of this manuscript.
  3. Page 2, ll 74 – 76 – specify change in relation to what?
  4. Page 2: “The effect of treatment on the bone mineral content is also a very important issue because 40% of bone mass is formed in childhood [18]. Both cancer and its treatment adversely affect bone formation, leading to osteoporosis and osteopenia in children with leukemia, solid and brain tumors, treated with drugs such as glucocorticosteroids, methotrexate, ifosfamide, cyclosporine, doxorubicin, and cisplatin [6, 19, 20]. - I propose as a separate paragraph and describe this important problem in more detail.
  5. Page 3: Assessment of the prevalence of malnutrition in children with solid tumors is difficult due to the limited number of research [11]. - reference from 10 years ago. Still no research?
  6. Page 3, ll 137-138: “It is also emphasized that pediatric patients with solid tumors have a higher risk of malnutrition, considering the MUAC, TSFT, and AMC indicators, compared to other methods” - Clarify the sentence. Is the risk of malnutrition dependent on the method?
  7. Table 1: add research references in the table.
  8. Page 6: I suggest changing subheading 4.1 from "definition" to e.g. Physiology of the ECS.
  9. Page 6, l 269: In the ECS there are two main types of receptors – CB1 and CB2 … - in new paragraph.
  10. Figure 1 – add a description of figure 1 in the body of text.
  11. Section 4.3 is definitely too modest. There are results of in vitro studies on the role of the ECS in pediatric cancers. It is worth presenting them.

Author Response

Dear Reviewer,

Reviewer 3 Report

The submitted review article aims at summarizing the possible correlation among nutritional status, gastrointestinal peptides, and endocannabinoid system in children with cancer. The topic is really interesting but experimental and clinical observations are far away to assess a possible link among nutritional status/gastrointestinal peptides, and endocannabinoid system in children with cancer.  In this respect the manuscript is too much preliminary for standard of the IJMS and mainly focused on the relevance of nutritional status in pediatric oncology than on the possible relationship between nutritional status and endocannabinoid system in children with cancer.

Furthermore, paragraph 4 is really limited in ECS description and functions and the The role of ECS in childhood cancer (par 4.3) seems like a speculation.

Additional minor points:

  • “page 1 abstract, The endocannabinoid system (ECS) is a mechanismwhose best-known function is the regulation of energy balance and food intake”: the ECS is a signaling system and not a mechanism; energy balance and food are only some of the large of functions of the ECS in biological systems

  • Be consistent with the use of abbreviations all over the text

Author Response

Point 1: “The topic is really interesting but experimental and clinical observations are far away to assess a possible link among nutritional status/gastrointestinal peptides, and endocannabinoid system in children with cancer. In this respect the manuscript is too much preliminary for standard of the IJMS and mainly focused on the relevance of nutritional status in pediatric oncology than on the possible relationship between nutritional status and endocannabinoid system in children with cancer.”

Response 1: We understand that data on the relationship between the nutritional status/ gastrointestinal peptides and endocannabinoids in chidren with cancer are very limited and therefore ofcourse we cannot show unambigous link between them. However, we wanted to present the current state of knowledge and indicate the need and direction of future research in this topic.

Point 2: “Paragraph 4 is really limited in ECS description and functions and the The role of ECS in childhood cancer (par 4.3) seems like a speculation.”

Response 2: We added Table 5. Characteristic of studies that assess cannabinoid anti-cancer activity in children, where we decribed research in this topic.

In paragraph 4 we focused mainly on the description of the ECS in the context of appetite regulation, as this is the main concern of the manuscript, therefore we only mentioned other functions of the ECS.

Point 3: “The endocannabinoid system (ECS) is a mechanism whose best-known function is the regulation of energy balance and food intake”: the ECS is a signaling system and not a mechanism; energy balance and food are only some of the large of functions of the ECS in biological systems

Response 3: We changed it for: “The endocannabinoid system (ECS) is a signaling system whose best-known function is the regulation of energy balance and food intake, but also plays a role in pain control, embryogenesis, neurogenesis, learning and regulation of lipid and glucose metabolism.”

Round 2

Reviewer 2 Report

In my opinion, the changes introduced increased the substantive value of the manuscript and will certainly have an impact on the citation of the publication.

Author Response

Dear Reviewer,

thank you so much for your valuable comments.

Reviewer 3 Report

The authors have significantly improved their review article, changing the focus of the article and including a large number of data on food intake/metabolism  in pediatric oncology.  The part dedicated to the ECS has been better structured; the link between pediatric oncology and ECS is still unravelled and remain a speculation (better presented with respect to the previous version of the manuscript.

Taken together, the revised version of the manuscript has suffuicient quality for publication.

Author Response

(The authors gave the same response as above.)
